# Study of the High-Efficiency Ejecting-Explosion EDM of SiCp/Al Composite

**DOI:** 10.3390/mi14071315

**Published:** 2023-06-27

**Authors:** Yu Liu, Jiawei Qu, Keguang Zhao, Xuanyuan Zhang, Shengfang Zhang

**Affiliations:** 1School of Mechanical Engineering, Dalian Jiaotong University, Dalian 116028, China; liuyu_ly12@126.com (Y.L.); qjw_12@126.com (J.Q.); zhangxuanyuan2016@outlook.com (X.Z.); 2Huazhong Institute of Electro-Optics, Wuhan 430233, China; wyb18742589157@163.com

**Keywords:** SiCp/Al composite, ejecting-explosion EDM, step wave, process law

## Abstract

SiCp/Al composites have excellent physical properties and are widely used in aerospace and other fields. Because of their poor machinability, they are often machined by non-traditional machining methods such as electrical discharge machining (EDM). In the process of EDM, due to the “shielding” effect of the reinforced particles of SiC, the local ejection force is low during the processing process, and it is difficult to throw the reinforced particles smoothly, which ultimately leads to a low material removal rate and poor surface quality. In this paper, a high–low-voltage composite ejecting-explosion EDM power supply is developed to explore the explosive effect of reinforced particles in the ejecting-explosion EDM process and the unique process law of the explosion process. The experiment platform uses self-developed CNC machining machine tools based on an ejecting-explosive EDM power supply, and the influence of a detonation-increasing wave on the processing of SiCp/Al composites with different volume fractions was studied by changing four factors: the open-circuit voltage difference, pulse current difference, pulse phase difference, and pulse width difference of the back wave behind the step front. The material removal rate and surface roughness were measured. The research results showed that the material removal rate could be increased to 164.63%, and the material surface roughness could be increased to 30.03% by adjusting the high and low pulse current difference from 1 A to 8 A. When the voltage difference between high and low wave (HLW) pulses increases from 40 V to 120 V, the material removal rate can be increased to 150.39%, and the material surface roughness can be increased to 20.49%. The material removal rate increases with the increase in pulse phase difference and open-circuit voltage difference. With the increase in peak current difference and pulse width difference, the material removal rate becomes faster at first and then slower. The surface roughness of materials increases with the growth of open-circuit voltage difference, peak current difference, pulse width difference, and pulse phase difference.

## 1. Introduction

SiCp/Al composites made by mechanical stirring and other methods can have the characteristics of high plasticity for metals and high brittleness for ceramics. SiCp/Al composite is widely used in aerospace, electronic packaging, high-speed rail, microwaves and other fields because of its low density, light weight, high specific strength, small thermal expansion coefficient, and other excellent characteristics [1,2,3]. However, SiC particles with high hardness and wear resistance are mixed in SiCp/Al composites, which seriously affects the service life of traditional machining tools and the surface quality of the workpiece. Therefore, non−traditional machining is considered to be one of the most effective machining methods for machining SiCp/Al composites [4,5]. Because EDM erodes materials through the electrothermal effect formed between two electrodes, it has no substantial cutting force and avoids the disadvantages of traditional tool machining, so EDM is considered to be an ideal method for machining SiCp/Al composites [6,7]. In the process of EDM, with the increase in machining depth, it is difficult for the debris to be thrown into the machining fluid; a large amount of debris becomes stacked at the bottom of the pit, which makes the inter-electrode discharge state unstable and the machining difficult. Therefore, the removal rate of materials becomes a key factor for machining.

Many domestic and foreign scholars achieve have improved the power supply for EDM. Chen et al. designed a micro-generator with multiple micro-resistance capacitors, which can generate high peak current and a short pulse, and greatly improve the material removal rate compared with a single RC circuit [8]. Chung et al. have conducted in-depth research on capacitive pulse power supplies to improve their energy utilization and reduce energy consumption. In the main circuit, an inductor is used instead of a current-limiting resistor, allowing excess energy to be returned to the power supply, greatly reducing energy loss. Additionally, the capacitor voltage can be adjusted by controlling the open-circuit voltage of the power supply to control its charging and discharging time [9]. Odulio et al. have developed an energy-saving pulse power supply for flyback EDM, in which the flyback transformer will act as a clamp winding when the load is an open circuit. When the reflected voltage at this clamp winding exceeds the input voltage, diodes D1 and D3 will conduct and will have a regenerative action that returns energy to the source [10]. Wang et al. proposed a high-frequency pulse power supply with switch controlled capacitor. The total energy charged into the discharge capacitor is controlled by the turn-on time of the switch at the primary side of the flyback converter. All discharge capacitors are charged to the same open-circuit voltage together. The switch connected in series with the discharge capacitor is turned on in high-frequency order, so that the discharge capacitor acts on the gap in turn to generate high-frequency discharge. It is characterized by low power loss, uniform discharge energy, and high discharge frequency [11]. Wu et al. proposed a pulse power supply based on a pulse generator, which uses a multi intersection stagger buck converter as the main circuit and uses high output voltage circuit to conduct gap breakdown. The multi-interleaving stagger buck converter controls the discharge current waveform to follow the given reference through high-frequency conversion mode operation and uses interleaving control technology to offset the discharge current ripple to a lower value. A high switching frequency is used in rough machining and a small discharge pulse is used in fine machining [12]. Yang et al. proposed a new bipolar pulse power supply, which uses advanced power electronics technology and has the characteristics of controllable discharge energy in micro EDM. It is possible to achieve a fixed discharge frequency with constant energy consumption in each discharge cycle, which improves processing efficiency [13].

Proper process parameters can significantly improve the efficiency of EDM, which has been studied by scholars at home and abroad. Tajdeen et al. predicted the influence of response parameters such as peak current, pulse width, and gap voltage on material removal rate (MRR) and surface roughness (SR) through a genetic algorithm. The experimental results show that the material removal rate increases with the increase in pulse width. The peak current has the greatest effect on the surface roughness, and the gap voltage has the least effect. The best input parameters of material removal rate are: peak current 4 A, pulse width 216.883 μs, The gap voltage is 30 V. The best input parameters of surface roughness are peak current 4 A and pulse width 125 μs. The gap voltage is 30 V [14]. Le conducted EDM on SKD61 steel under finishing and semi-finishing. The experimental results show the peak current (1 A) and pulse width (16 μs) The surface roughness and hardness are better than the high peak current (4 A) and long pulse width (200 μs) results by [15]. Jose and Paul analyzed the EDM test by Taguchi using the orthogonal method and found that the discharge gap is in direct proportion to the gap voltage, an important parameter that affects the surface roughness and material removal rate of the processed material. The comparison test found that the gap distance of 0.5 mm is the best [16].

Khan et al. conducted EDM on shape-memory alloy (SM) and found that the power supply voltage caused the most obvious damage to the surface of the machined material [17]. Sivakumar et al. used peak current, pulse width, and pulse voltage as input parameters to EDM OHNS steel, and found that a lower peak current would produce an ideal material surface roughness [18]. Ganapathy et al. exploited the Taguchi experiment design and the use of GRA was combined with an L9 orthogonal array. A set of process parameters with a peak current of 21 A, a pulse width of 1000 μs and a voltage of 1.2 kg/cm^2^ was obtained by optimizing process parameters such as peak current, pulse width, and open-circuit voltage, using EN-31 steel as the object of study, resulting in a material removal rate of 80.52%ni [19]. Chandramoulis et al. studied the Taguchi method to optimize the process parameters of copper tungsten electrode steel-processing materials. The research results show that the pulse on time and discharge current have significant effects on the material removal rate and surface roughness, while the pulse off time has the least effect, which has been verified by experiments [20].

In order to improve the efficiency of the reinforcing particles thrown from the molten pool and explore the influence of the ejecting-explosion EDM processing on the surface roughness of workpiece materials under different electrical parameters, this paper introduces a self-made EDM power supply for processing, and the influence of the open-circuit voltage difference, peak current difference, pulse phase difference and pulse width difference on the material removal rate and surface roughness of the ejecting-explosion EDM power supply is studied experimentally, which is necessary to improve the EDM efficiency. However, the influence of the open-circuit voltage difference, peak current difference, pulse phase difference, and pulse width difference of the ejecting-explosion EDM power supply on the material removal speed and surface roughness is still unclear, so it is particularly important to master the influence of the above parameters on EDM.

## 2. Experiment on Ejecting-Explosion EDM

The ejecting-explosion EDM power supply is proposed for particle-reinforced metal matrix composites (PRMMCs), which are difficult to machine. It adopts a high- and low-voltage combined ejecting-explosion method. In order to achieve the purpose of high- and low-voltage composite processing, a control pulse signal with timing logic is generated by the control-pulse-generation device, and the power switch is turned on and off through the corresponding driving circuit, so that the high voltage and low voltage are loaded into the discharge gap according to the control logic to complete the combination of high- and low-voltage waveforms, as shown in Figure 1. First, the first level of low-voltage pulse has the same function as the ordinary spark pulse power supply. It breaks through the discharge gap and forms a discharge channel, which plays a role in the corrosion of PRMMCs. Due to the great difference in the physical properties of the metal matrix and the reinforced particles, the melting time and required energy of the two materials are not the same. In order to avoid the inefficient process of increasing the discharge energy to melt the reinforced particles and directly throw out the reinforced particles in the form of a solid with molten metal entrainment, after the first low-voltage pulse is completed, the second high-voltage pulse is introduced. This pulse has a higher peak voltage and current, which releases greater energy at the moment of throwing out the material, obtaining greater discharge explosive force. The reinforced particles in the discharge molten pool are thrown out with the molten metal entrainment in the form of a solid, so as to achieve the purpose of the efficient removal of materials.

### 2.1. Experimental Materials

In this paper, SiCp/Al composites with different volume fractions of SiC particles are used for comparative experiments. Before the experiment, the SiCp/Al composite material with each volume fraction was cut into 20 mm × 20 mm × 10 mm rectangular squares by wire electrical discharge machining, and the specimen was polished from low to high using different mesh sandpaper to remove the oxide layer on the surface of the workpiece, and then the workpiece was cleaned with acetone solution to remove the attached impurities, so as to ensure the reliability of the experimental data and reduce the measurement error. The matrix materials are 6061 aluminum alloy, and the volume fractions of SiC are 20%, 30%, 45%, and 65%. The SiCp/Al composites are mainly prepared by powder metallurgy; after mixing aluminum powder and silicon carbide powder, they are pressed into blanks under vacuum and high pressure, and finally formed SiCp/Al composites after heat treatment. The diameter of 20% volume fraction SiC particles is 7 μm. Both 30% and 45% volume fraction SiC particles have a diameter of 14 μm, and 65% volume fraction SiC particles a diameter of 20 μm. The comparison of the main material performance indexes is shown in Table 1.

### 2.2. Ejecting-Explosion EDM Experiment Device

The experiment is based on the EDM experiment platform, including the self-developed CNC machine tool developed and the self-developed ejecting-explosion EDM power supply. The machine tool is controlled by the CNC software. The three axes drive the ball screw through the stepping motor to achieve transmission, which can achieve X, Y, and Z axis CNC positioning control. The working parameters of the machine tool are shown in Table 2.

The basic parameters of the step wave can be set for the electric ejecting-explosion power supply through the pulse-signal-generation circuit. The machining waveform required to generate power from the pulse generator. When the processing waveform is output from the signal generation circuit, it enters the power amplification circuit. The front end of the power amplifier circuit is an optocoupler isolation circuit. To prevent the high voltage of the rear-end circuit from affecting the operation of the front-end circuit, the optocoupler chip is used for electrical isolation and noise reduction. The signal is transmitted to the MOSFET drive circuit. The MOSFET drive chip is responsible for amplifying the signal generated by the generation circuit to drive the power MOSFET to respond quickly. The high- and low-circuit MOSFET drive signals, respectively, control the “on” and “off” of the series and parallel MOSFET switches, thus controlling the high and low voltage loaded in the discharge gap. Finally, the discharge-gap-detection part provides real-time feedback on the processing status to ensure the stable operation of the processing process. The actual processing is shown in Figure 2.

### 2.3. Experimental Design Scheme

The influence of the waveform parameters of the different front and rear of the step wave on the ejecting-explosion EDM effect needs to be studied experimentally. In this paper, the open-circuit voltage difference, peak current difference, pulse phase difference, and pulse width difference of the front and rear of the step wave are changed to study the influence of the rear edge detonation wave on the machining process of SiCp/Al composites. Four groups of continuous pulse machining experiments under the same parameters were carried out to study the influence of the difference of the parameters of the above step wave on the process. The experiment processing time is 1 min, the processing speed is set to about 1 mm/min, the machining pattern is all ejecting-explosion mode, the workpiece material is not broken, and the rest of the experiment parameters are set as shown in Table 1 and Table 3.

### 2.4. Experiment Index and Method

(1)Machining removal rate

The material removal rate (MRR mm^3^/min) of EDM reflects the processing speed. It refers to the amount of the workpiece material removed per unit of time under certain processing conditions, which is directly related to the processing time and processing efficiency of the workpiece. It is usually divided into mass removal rate and volume removal rate. At present, the volume removal rate is widely used in practice, which can more intuitively reflect the removal of the material. In this paper, the volume removal rate is used, and the formula is expressed as follows:(1)$$V_{s}=\frac{V}{t}=\frac{\Delta m}{\rho t}$$
where *V*_s_ is the volume removal rate (mm^3^/min), *V* is the volume of workpiece removal (mm^3^), *t* is the processing time (min), Δ*m* is the mass of workpiece removal (g), and *ρ* is the workpiece material density (g/mm^3^).

The Alicona automatic zoom three-dimensional surface topography instrument is used to measure the surface of the processed workpiece. The three-dimensional surface topography instrument can scan the whole surface of the object vertically by changing the depth of focus. According to the surface condition of the three-dimensional results, the plane is selected, and the plane is used as the reference plane. The depression and the protrusion volume of the set plane are measured, and the built-in difference analysis method in the three-dimensional surface topography instrument is supplemented to measure the volume accurately (by comparing the difference of the three-dimensional morphology front and rear processing, the volume of removal and protrusion is obtained).

(2)Machining roughness of pit bottom

Because SiCp/Al composites are not uniform materials, there are SiC reinforced particles in them. If the reinforced particles can be effectively removed, surface protrusions are greatly reduced, and the roughness after processing is also improved. Therefore, the surface quality after processing can be seen macroscopically by measuring the roughness of processing. In this paper, a three-dimensional topography instrument is used to divide and measure the bottom area of the processing pit, so as to reduce the influence of the edge-loss effect caused by the smooth processing of the electrode and reduce the height error of the peak and valley caused by the concave at the bottom, and then the average evaluation is carried out to obtain the comprehensive processing roughness.

## 3. Analysis on Influencing Factors of Material Removal Speed in Ejecting-Explosion EDM

### 3.1. Influence of Voltage Difference of HLW

In order to explore the effect of the open-circuit voltage difference between high- and low-voltage waveform loading on the material removal rate of SiCp/Al composites by EDM, different open-circuit voltage differences were selected for EDM continuous pulse machining experiment. Experimental conditions are as follows: high- and low-voltage waveform peak current of 3 A and 5 A, HLW pulse width of 20 μs, low-voltage wave open-circuit voltage is set to 80 V, and high-voltage wave open-circuit voltage is set to 120 V, 140 V, 160 V, 180 V, 200 V. The experiment results show the influence curves of different open-circuit voltage differences on the material removal volume and rate in Figure 3.

The experimental results shown in Figure 3 show that the removal rate of SiCp/Al composites with different volume fractions increases with the increase in the open-circuit voltage difference between high- and low-voltage waveform. According to the slope of the curve, it can be seen that with the increase in the open-circuit voltage difference between high- and low-voltage waveform, the growth rate of the removal volume slows down, and the influence of the voltage difference between high- and low-voltage waveform on the removal volume gradually decreases.

This law occurs because as the open-circuit voltage difference between high–low-voltage waveforms increases, the energy released by the pulse power supply between the discharge gaps at the same time also increases, and the material removal volume per unit time also increases. At the same time, the increase in the open-circuit voltage of the high-voltage wave also increases the discharge gap. Combined with the strong explosive force generated by the rear edge, the debris can be discharged more effectively. However, with the increase in the open-circuit voltage value of the high-voltage circuit, the discharge gap becomes larger, the release space of metal steam and working fluid steam generated at high temperature becomes larger, and the pressure increase generated with the increase in energy becomes smaller. Therefore, the explosion force brought by the pressure release decreases with the increase in the voltage difference. At the same time, the volume of removal increases with the increase in the voltage difference. There is more and more debris generated between the poles. The reinforced particles of SiC and the molten product of the aluminum matrix affect the discharge effect. Under the influence of various factors, when the open-circuit voltage difference of the high–low-voltage waveform loading increases to a certain value, the removal rate of the workpiece material and the throwing effect of the debris reach the optimal situation. If the open-circuit voltage difference continues to increase, the effect on the material throwing gradually decreases, and the removal volume gradually tends to a stable value. At the same time, the excessive discharge gap leads to the instability of the machining process, which affects the machining speed and quality. Therefore, combined with the experimental results and theoretical analysis, it can be seen that the high–low-voltage waveform loading open-circuit voltage difference is better processed at 80 V to 100 V.

Due to the poor conductivity of the reinforced particles of SiC, the “shielding” effect is generated during the processing, which blocks the formation of the discharge channel. Therefore, with the increase in the volume fraction of the reinforced particles of SiC, the removal volume is less under the same electrical parameters. However, when the open-circuit voltage difference of the high–low-voltage waveform loading reaches a certain degree, the removal volume of the SiCp/Al composite with a volume fraction of 45% is higher than that of the SiCp/Al composite with a volume fraction of 30%. This is because the ratio of SiC particles and aluminum matrix in the 45% SiCp/Al composite material at 60 V voltage can just make the molten aluminum matrix with SiC particles thrown out, which is larger than the low volume fraction of molten aluminum. However, the shielding effect of reinforced SiC particles in 65% SiCp/Al composites is too obvious, which makes the material achieve less explosive force. Therefore, it is impossible to remove more volume under high-voltage energy, as for 45% SiCp/Al composites. Therefore, it still follows that with the increase in the volume fraction of reinforced particles of SiC, under the same electrical parameters, the removal volume is less regular.

### 3.2. Influence of Peak Current Difference of HLW

In order to investigate the effect of the peak current difference of high–low-voltage waveform loading on the material removal rate of SiCp/Al composites by EDM, different peak current differences were selected for EDM continuous pulse machining experiment. Experimental conditions are as follows: high–low-voltage waveform loading open-circuit voltage are 80 V and 100 V, respectively; HLW pulse width of 20 μs; low-voltage peak current is set to 4 A; high voltage peak current is set to 5 A, 6 A, 8 A, 10 A, 12 A; and the difference between the peak currents of high–low-voltage waves is 1 A, 2 A, 4 A, 6 A, 8 A. The experimental results show the influence curves of different peak current differences on the material removal volume and rate in Figure 4.

The experimental results shown in Figure 4 show that the material removal rate of SiCp/Al composites with different volume fractions increases with the increase in the peak current difference of high–low-voltage waveform loading. However, the curve trends of SiCp/Al composites with low volume fractions of 20% and 30% are very different from those of SiCp/Al composites with high volume fractions of 45% and 60%. It can be seen that the removal volume of SiCp/Al composites with low volume fraction decreases with the increase in the peak current difference of high–low-voltage waveform loading. However, the removal rate decreases significantly after 6 A, while the material removal rate of SiCp/Al composites with high volume fraction increases with the increase in peak current difference between high–low-voltage waveforms.

Although the removal rate of SiCp/Al composites with a low volume fraction increases with the peak current difference of high–low-voltage waveform loading, the increase in removal rate decreases. The removal rate decreases after the peak current difference of high–low-voltage waveform loading reaches 6 A. This is because the peak current of the high-voltage circuit continues to increase, which leads to an increase in discharge energy, increasing the number of charged ions in the discharge channel. A larger current density also increases the kinetic energy of charged particles. The temperature of the discharge channel, the workpiece, and the electrode surface increases under the bombardment of charged particles, and the volume of the melted and vaporized aluminum matrix material increases. After the material is vaporized, the volume becomes larger, the discharge gap pressure becomes larger, and the pressure is released. When the pressure is released and the discharge energy brings a stronger explosive force, the more melted metal is thrown, and the processing efficiency is higher. However, the discharge gap remains unchanged. In the case of an increase in the volume of material removal, a stronger explosive force can discharge more debris, and more residual debris remains between the poles. In the continuous processing process, as the retention accumulates, it affects the processing rate of removal, so the increase in removal rate gradually decreases. When the current difference is too large, the SiC particles are also melted, which leads to a decrease in the material removal rate; in addition, when the current difference is too large, the peak current of the high-voltage circuit is too large, which leads to the charged ion movement being too violent, resulting in more complex reactions between the poles, resulting in poor chip removal and high discharge channel temperature. It is difficult to eliminate ionization, so excessive peak current difference reduces the workpiece removal rate. The SiCp/Al composites with a high volume fraction have a higher proportion of reinforced particles of SiC. The larger the peak current of the high-voltage circuit, the more generated charged particles bombard a higher proportion of reinforced particles of SiC, while the SiC particles have a “shielding” effect on the discharge gap, so the reaction intensity of the charged particles is relatively lower. At the same time, the thermal conductivity of SiC is much larger than that of the aluminum matrix, so the temperature in the discharge channel is relatively lower, and the deionization situation is improved. Therefore, after the current difference of 6 A continues to increase, it still shows a trend of greater removal rate, but the increase in the removal rate also decreases.

### 3.3. Influence of Peak Current Difference of HLW

In order to explore the effect of the pulse width difference of high–low-voltage waveform loading on the material removal rate of SiCp/Al composites by EDM, different pulse width differences were selected for the EDM continuous pulse machining experiment. Experiment conditions are as follows: high–low-voltage waveform loading open-circuit voltages are 80 V and 160 V; peak current is 3 A and 5 A; low-voltage wave pulse width is set to 35 μs, 30 μs, 20 μs, 10 μs, 5 μs; and high-voltage wave pulse width is set to 5 μs, 10 μs, 20 μs, 30 μs, 35 μs. The influence curves of different pulse width differences on the material removal rate according to the experiment results are shown in Figure 5.

The experimental results shown in Figure 5 show that the removal rate of SiCp/Al composites with different volume fractions increases first and then decreases gradually with the increase in pulse width difference because the front wave electrical parameters have a good effect on the removal of SiCp/Al composites. Due to the particularity of SiCp/Al composites, when the workpiece material is removed, there is a reinforced particle of SiC free in the discharge gap, which affects the normal discharge operation of the machining, so the surge energy is provided by the trailing edge high-voltage wave. The pressure difference caused by the formation of energy mutation and the stronger explosive force enhance the throwing of particles of SiC from the workpiece and render the discharge smoother. With the increase in the pulse width of the rear edge high-voltage wave, although the rear edge high-voltage wave can melt more materials and generate greater explosive force, the pressure action time brought by the energy surge moment come relatively forward with the increase in the pulse width difference. Therefore, there is more discharge formed in the discharge gap. At this time, the action time of the high-voltage waveform accounts for more, and the energy generated between the poles is excessive, which also renders the effect of deionization between the poles worse. Therefore, continuous pulse machining leads to a decrease in the removal rate. For the SiCp/Al composite with a volume fraction of 65%, the high-voltage waveform has a long action time; more energy is released, and the reaction is more intense. Due to the presence of more reinforced particles of SiC, the thermal conductivity is greater, the interpolar energy dissipates faster, and the deionization effect is unaffected. When the high-voltage wave action time is longer, the 65% SiCp/Al composite volume fraction is more significant.

### 3.4. Influence of Phase Difference of HLW

To explore the effect of the pulse phase difference of high–low-voltage waveform loading on the material removal rate of SiCp/Al composites by EDM, different pulse phase difference values were selected for EDM continuous pulse machining experiment. Experiment conditions are as follows: high–low-voltage waveform loading open-circuit voltages are 80 V and 100 V; peak current is 3 A and 5 A; HLW pulse width accounted for 20 μs; high-voltage wave pulse phase is set to 0%, 25%, 50%, 75%, 100%; and the rest of the high-voltage wave pulse is not set at the low-voltage wave pulse. The experiment results show the influence curves of different pulse width differences on the material removal rate in Figure 6.

The experimental results shown in Figure 6 show that when the phase difference between the high–low-voltage waveforms becomes larger, the removal rate decreases first and then increases. Because the current high-voltage wave is first loaded in the discharge gap, the high-voltage wave can break through the discharge gap faster, thus forming the discharge channel more quickly. When the high-voltage wave is loaded, the low-voltage wave is loaded in the discharge gap, and the energy is directly released in the discharge gap. Although there is no energy surge in the machining process, the faster breakdown gap makes the energy utilization rate higher, so the removal volume effect is better. When the phase difference becomes slightly larger, the low-voltage wave is loaded for a short period of time, and the gap is not broken down. The high-voltage wave is loaded into the discharge gap, thereby breaking the voltage. The subsequent inter-electrode energy changes are the same, but some of the low-voltage waves release waste energy, so the removal volume is reduced at this time. When the phase difference between the high–low-voltage waves is larger, the low-voltage wave breaks down the discharge gap, and the high-voltage wave is loaded into the discharge gap at this time. At this time, there is a surge of energy, so greater energy release forms a stronger explosive force. At the same time, the instantaneous high temperature vaporizes more metals and working fluids at the same time, forming a greater vapor pressure. The simultaneous action of the two can strip more reinforced particles of SiC and the aluminum matrix melts from the workpiece and discharge gap, so the effect of etching volume is more obvious at this time. However, after the subsequent high-voltage wave loading is completed, the low-voltage wave continues to load. At this time, most of the melting products on the surface of the workpiece have been discharged, and the effect of subsequent low-voltage wave removal is not so obvious. With the maximum phase difference of high–low-voltage waveforms, when the low-voltage wave is fully loaded, the high-voltage wave is loaded again. At this time, the action time of the low-voltage wave is longer, so it has relatively more melts formed on the workpiece material. At this time, the high-voltage wave provides an instantaneous energy surge, and the explosive force and steam pressure are enhanced. More melts and reinforced particles of SiC are peeled off to process the workpiece, so the processing removal volume is the largest at this time, and the removal effect also reaches the best state.

## 4. Analysis of Factors Affecting Surface Roughness of Materials in Ejecting-Explosion EDM

The surface quality of the material after processing has a crucial influence on the service life and performance of the material. Under different processing conditions, the surface morphology after EDM is different. In order to explore the effect of different processing parameters on the surface roughness of the material, the SiCp/Al composite materials with different volume fractions after processing were taken, and the surface morphology at the center of the bottom of the discharge pit was observed and compared.

### 4.1. Effect of High–Low-Voltage Difference on Surface Roughness and Surface Quality

In order to explore the influence of open-circuit voltage difference on the surface roughness and surface quality, different open-circuit voltage differences are selected for comparative experiments with EDM. The parameters of the open-circuit voltage difference are the same as those of the previous section. According to the experiment results, the influence curve of open-circuit voltage difference on surface roughness is drawn in Figure 7.

The experimental results shown in Figure 7 show that the surface roughness increases with the increase in open-circuit voltage. The roughness of SiCp/Al composites with low volume fraction is larger than that of 60 V when the voltage difference is 40 V. Because the discharge effect is poor and the discharge is unstable at low voltage differences, the roughness is larger under low-voltage parameters. Due to the shielding effect of SiC particles, the processing intensity decreases with the increase in volume fraction under the same waveform parameters, so the corresponding roughness decreases with the increase in volume fraction. However, the roughness of 45% volume fraction SiCp/Al composites is greater than that of 30% volume fraction SiCp/Al composites when the voltage difference is greater than 80 V because 45% SiCp/Al composites have more reinforced particles thrown out of the matrix by explosive force under this waveform parameter, and the pits left after throwing out the particles make the surface fluctuate more. Therefore, the roughness is greater.

Combined with the micro-morphology of the surface of the low-volume-fraction and high-volume-fraction SiCp/Al composites under different voltage difference parameters shown in Figure 8, it can be seen that the low-volume-fraction SiCp/Al composites have a high proportion of aluminum matrix after EDM, so the surface of the SiCp/Al composites with a relatively high volume fraction retain more metal-melting products, resulting in relatively more pits and depressions on the surface, while the high-volume-fraction SiCp/Al composites have a relatively flat surface after machining. Therefore, the surface roughness of SiCp/Al composites with a low volume fraction is often higher than that of those with a high volume fraction. As the voltage difference continues to increase, it can be seen that the cracks remaining on the surface of the workpiece gradually become larger and wider. When the voltage is continuously increased, the energy released between the poles also continue to increase. In the process of EDM, according to the loading and interval of energy release, high temperature and low temperature are continuously interspersed in the workpiece removal position. This process causes thermal stress inside the workpiece material. When the stress is low, it exists on the surface of the workpiece in the case of residual stress. As the energy release continues to increase, the alternating range of temperature also increase, resulting in greater thermal stress. When the voltage difference is larger, the temperature change is more complex, so the crack becomes larger and wider. At the same time, due to the better peeling effect of the reinforced particles under high-voltage explosion, the particles remain pits on the surface after being peeled off. At this time, the residual holes on the surface also increase. Although the chip removal effect is better under high explosive force, the surface roughness continues to increase due to the increase in the number of larger cracks and holes on the surface and the irregular condensation caused by the stronger processing reaction.

### 4.2. Effect of High–Low Current Difference on Surface Roughness and Surface Quality

In order to explore the influence of peak current difference on surface roughness and surface quality, different peak current differences were selected for the EDM comparison experiment. According to the experiment results, the influence curve of the peak current difference on surface roughness was drawn.

The experimental results shown in Figure 9 show that the roughness of the four volume fractions of SiCp/Al composites increases with the increase in the current difference. The increase in the peak current of the high-voltage wave increases the number of charged ions; the volume of micro-pit removal is larger, and the roughness continues to increase. However, the roughness of the low volume fraction increases sharply at a current difference of 6 A. At this time, the peak current of the high-voltage circuit is too large, which leads to the movement of charged ions being too violent, so that the low-melting-point aluminum matrix with more content is melted. At this time, the pole spacing changes less, resulting in more complex pole reactions, resulting in poor chip removal and high discharge channel temperature, which makes it difficult to deionize. The processing effect is poor, so the roughness becomes larger. Compared with SiCp/Al composites with low volume fraction, SiCp/Al composites with high volume fraction contain more reinforced particles of SiC, so the reaction is not so intense relatively. With the increase in current difference, the explosive force leads to more reinforced particles thrown from the matrix, and the excess energy melts the aluminum matrix. Due to the greater energy release, the explosive force is also greater, and the roughness also increases.

Combined with the micro-morphology of low-volume-fraction and high-volume-fraction aluminum-based silicon carbide surface processing under different current difference parameters shown in Figure 10, it can be seen that after EDM, SiCp/Al composites with a relatively high volume fraction on the surface remain more metal-melting products, so the surface roughness of a low volume fraction is greater than that of a high volume fraction workpiece under the same parameters. With the continuous increase in the current difference, it can be seen that the holes on the surface of the workpiece gradually increase with the continuous increase in the current difference. The energy released between the poles continues to increase. When the current difference is small, the increase in the explosion energy at the rear edge is small, so the effect of particle removal is poor. The low-volume-fraction aluminum matrix accounts for a relatively high proportion, and the particles remove fewer holes. The remelting of the molten matrix metal covers the holes, so the number of holes is less than that of the high volume fraction. However, as the current difference becomes larger, the number of charged particles between the poles increases, and the discharge density becomes larger. At this time, the heat between the poles is more intense, the particle discharge effect is better, and the melting volume of the aluminum matrix is also more. The debris is discharged from the discharge gap under the action of greater explosive force brought by the explosion, and relatively less heavy metal condensate remains on the machined surface. At this time, the number of holes on the machined surface increases. When the current difference is too large, due to the same discharge gap, the movement of charged particles between the poles is too intense, which easily causes abnormal discharge, and the discharge effect of debris becomes worse. Therefore, although the surface holes multiply, the metal condensate is also more covered on the surface, and the number of surface metal-melting spheres increases. At the same time, because more reinforcing particles remain on the surface, they are wrapped in the metal recondensation layer to form internal stress, and the strength of the metal recondensation layer is weak, which leads to more small, dense, square-like cracks on the surface.

### 4.3. Effect of High–Low Pulse Width Difference on Surface Roughness and Surface Quality

In order to explore the influence of pulse width difference on surface roughness and surface quality, different pulse width differences are selected for the EDM comparison experiment. The parameters of pulse width difference are the same as those of the previous section. According to the experiment results, the influence curve of pulse width difference on surface roughness is shown in Figure 11.

The experimental results shown in Figure 11 show that the roughness of the four kinds of SiCp/Al composites increases with the increase in the pulse width difference. When the pulse width of the high-voltage wave becomes larger, the time of its action on the pole is more. The larger the energy released at a single time, the larger the size of the single micro-pit formed. The surface roughness generated after the superposition of micro-pits after multiple pulse removal is higher, and the roughness continues to decrease with the increase in volume fraction because of the “shielding” effect of SiC reinforced particles. The removal effect of 65% volume fraction SiCp/Al composites is better under high voltage pulse. When the pulse width difference is large, the removal volume is larger. When the larger volume of SiC reinforced particles is thrown out of the matrix, and some of the unpeeled reinforced particles after processing make the bottom surface fluctuate more, the roughness is higher than that of a relatively low volume fraction of 30% and 45%.

Combined with the micro-morphology of low-volume-fraction and high-volume-fraction SiCp/Al surface processing under different pulse width difference parameters shown in Figure 12, it can be seen that after EDM, the surface of relatively high-volume-fraction SiCp/Al composites retains more metal-melting products, so the surface roughness of the low-volume-fraction is greater than that of the high-volume-fraction workpiece under the same parameters. When the pulse width difference is small, the loading time of the rear high-voltage wave is relatively late, the volume of the molten metal of the front low-voltage wave is limited, and the explosive force brought by the surge energy can effectively discharge the debris from the discharge gap. At this time, the processed surface has fewer re-condensates and more surface holes. As the pulse width difference becomes larger, the loading time of the rear edge high-voltage wave becomes longer, and the time of the energy surge is earlier. Although the energy released along the high-voltage wave is large, there is no sudden explosion force because there is no surge moment, so the removal effect of the workpiece material melted by the subsequent high-voltage wave decreases instead. Therefore, it can be seen that there are more residual melts on the surface of the workpiece, and even some of the holes formed by the peeling of the workpiece at the surge moment are also covered by the condensate.

### 4.4. Effect of HLW Phase Difference on Surface Roughness and Surface Quality

In order to explore the influence of pulse phase difference on surface roughness and surface quality, different pulse phase differences are selected for the EDM comparison experiment. The processing parameters of the pulse phase difference are consistent with the previous section. According to the experiment results, the influence curve of the pulse phase difference on the surface roughness is drawn.

The experimental results shown in Figure 13 show that the roughness of SiCp/Al composites with four volume fractions increases with the increase in phase difference. When the high-voltage pulse is first loaded in the discharge gap, the breakdown time is shorter, the low-voltage wave is loaded in the back, the released energy is less, there is no state of energy surge, the reaction is more moderate, the energy utilization rate is better, the removal effect is better, and the machined surface is better. At the same time, the roughness is lower, and when the phase difference becomes larger, there is an energy surge, each pulse releases energy to make the explosive force greater, and the pole reaction is more intense. As the load time of the low-voltage wave increases, the explosive force generated by the energy surge during the high-voltage wave loading throws more enhanced particles and debris. The aluminum matrix of SiCp/Al composites with low volume fraction accounts for a relatively high proportion. When the explosive force effect is the same, its influence on the re-solidified aluminum matrix after melting continues to weaken. Therefore, the subsequent roughness of the low volume fraction tends to be minor, while there are more reinforced particles in the high volume fraction. When the high-voltage wave action occurs later, the discharge effect of reinforced particles and melts is better. When more reinforced particles are thrown from the workpiece, the residual pits and energy release are continuously superimposed, and the roughness of the machined bottom surface also increases.

Combined with the micro-morphology of low-volume-fraction and high-volume-fraction aluminum-based silicon carbide surface processing under different phase difference parameters of front and back waveforms shown in Figure 14, it can be seen after the EDM of low-volume fraction SiCp/Al composites that SiCp/Al composites with a relatively high volume fraction on the surface retain more metal-melting products, so the surface roughness of a low-volume-fraction is greater than that of a high-volume-fraction workpiece under the same parameters. When the phase difference is small and the high-voltage pulse is first loaded in the discharge gap, the breakdown time is shorter, the low-voltage wave is loaded at the back, the subsequent processing reaction is more moderate, and there is no energy surge. At this time, the processing surface is relatively flat and there are fewer holes, which is similar to the surface of ordinary EDM. As the phase difference becomes larger, when the low-voltage wave breaks through the gap and the high-voltage wave is loaded in the gap, an energy surge occurs. The pressure in the gap becomes larger at the moment of the surge, and the explosive force increases, which makes the enhanced particles peel off from the workpiece. At the same time, the explosive force better discharges the debris, so there are more holes on the processing surface. When the subsequent low-voltage wave is loaded again, the subsequent debris between the poles is freed, and the subsequent low-voltage wave itself generates less explosive force. Therefore, some debris remains on the final machined surface, thus re-condensing on the workpiece surface. When the high-voltage wave is completely placed and loaded between the poles in the low-voltage wave, the workpiece is first fully melted by the low-voltage wave, and then most of the material melted by the front energy is thrown out of the discharge gap as the energy surges, and then the high-voltage wave is loaded. During the process, it continues to melt and produces explosive force for discharge. Due to the greater energy density at this time, the removal reaction is more intense, so the final machined surface retains a large number of holes and crack-like morphologies, and it can be seen that relatively fewer recondensed melting products are attached to the surface.

## 5. Conclusions

In this paper, for SiCp/Al composites with four volume fractions, the ejecting-explosion EDM experiment was carried out using the ejecting-explosion EDM power supply. The effects of the open-circuit voltage difference, peak current difference, pulse width difference, and pulse phase difference on the removal rate and surface roughness of SiCp/Al composites with different volume fractions were studied by single-factor experiments, as follows:(1)The influence of waveform parameters on the material removal rate of SiCp/Al composites with different volume fractions was studied. Taking 45% SiCp/Al composites as an example, the results show that when the open-circuit voltage difference increases from 40 V to 120 V, the removal rate increases by 150.39%, but the removal rate slows down. When the peak current difference increases from 1 A to 8 A, the removal rate increases by 164.63%. When the pulse width difference increases from −30 μs to 30 μs, the removal rate increases by 2.89%. When the pulse phase difference increases from −100% to 100%, the removal rate increases by 71.41%. The material removal rate increases with the increase in pulse phase difference and open-circuit voltage difference, and first increases and then decreases with the increase in peak current difference and pulse width difference.(2)The influence of waveform parameters on the surface roughness of SiCp/Al composites with different volume fractions was studied. Taking 45% volume fraction SiCp/Al composites as an example, the results show that when the open-circuit voltage difference increases from 40 V to 120 V, the surface roughness increases by 20.49%. When the peak current difference increases from 1 A to 8 A, the surface roughness increases by 30.03%. When the pulse width difference increases from −30 μs to 30 μs, the surface roughness increases by 29.31%. When the pulse phase difference increases from −100% to 100%, the surface roughness increases by 41.06%. The surface roughness of the material increases with the open-circuit voltage difference, peak current difference, pulse width difference, and pulse phase difference.

## 6. Innovation and Prospects for Future Research

Due to the difficulty of SiCp/Al composites, ordinary EDM struggles to achieve the purpose of efficient continuous processing, so this paper introduces a ejecting-explosion EDM power supply to achieve the efficient processability of SiCp/Al composite EDM processing, and obtains the unique process law of SiCp/Al composites. Ejecting-explosion EDM can play an important role in the aerospace field. Fighter hatch covers, aircraft hydraulic components, Mars rover wheels, etc., are inseparable from the efficient processing of SiCp/Al composite materials.

Although this experiment can be used to improve the processing environment and improve the processing efficiency of SiCp/Al composites, the following disadvantages were found in the experiment:(1)Due to the limited research time, the experiment only studies the electrical parameters of the ejecting-explosion EDM process; however, the next step can be to conduct the experimental research on the SiCp/Al composite material for non-electrical parameters such as different electrodes and different working fluids, and establish a more complete database.(2)After the completion of the ejecting-explosion EDM experiment, only the surface quality and material removal rate of the workpiece were analyzed. If the state and elements of the debris after processing of SiCp/Al composites with different volume fractions were analyzed at the same time, the removal method of the ejecting-explosion EDM could be comprehensively explained.

## Figures and Tables

**Figure 1 micromachines-14-01315-f001:**
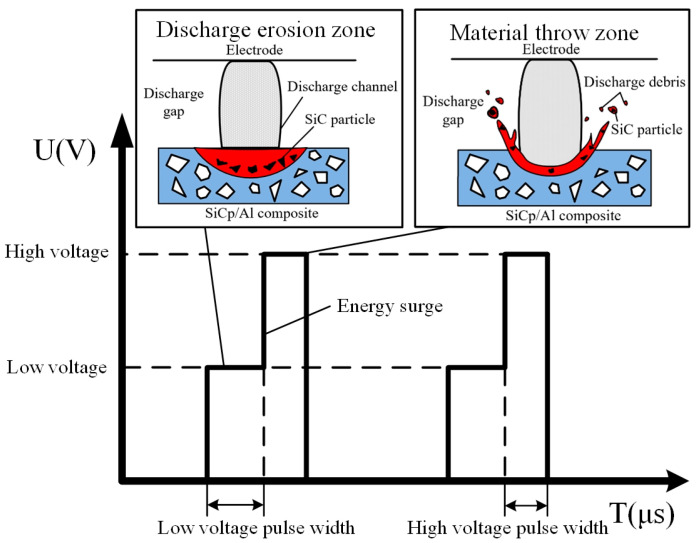
High- and low-voltage composite waveform.

**Figure 2 micromachines-14-01315-f002:**
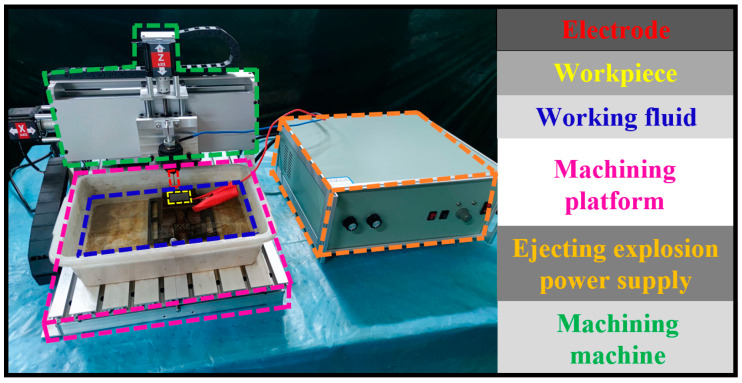
Live picture of EDM.

**Figure 3 micromachines-14-01315-f003:**
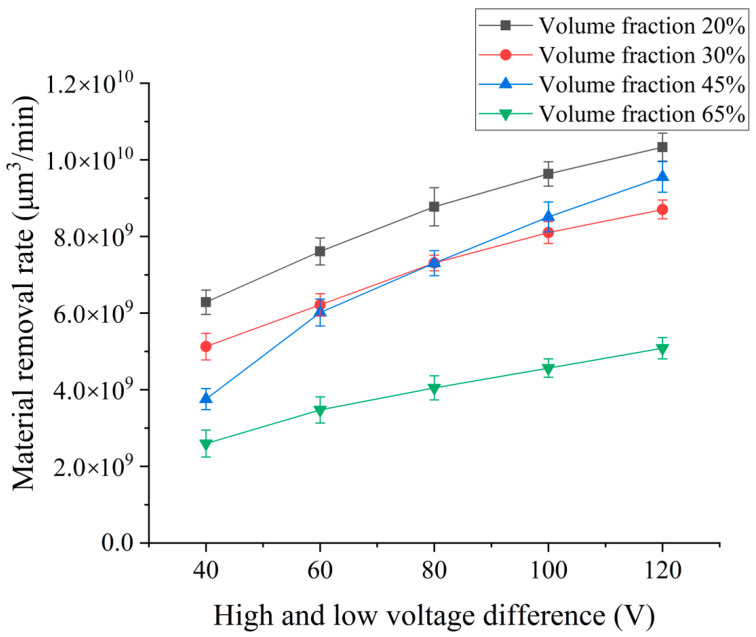
Effect of pulsed open-circuit voltage difference on removal.

**Figure 4 micromachines-14-01315-f004:**
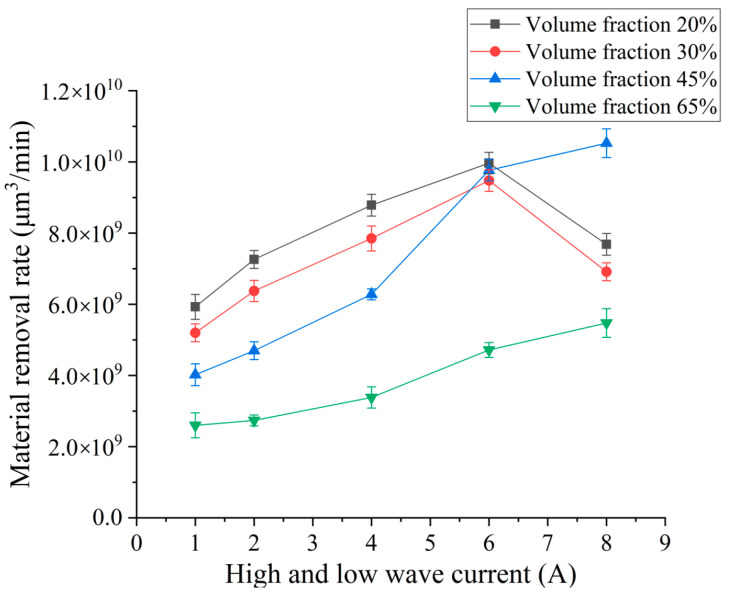
Effect of pulsed peak current difference on removal.

**Figure 5 micromachines-14-01315-f005:**
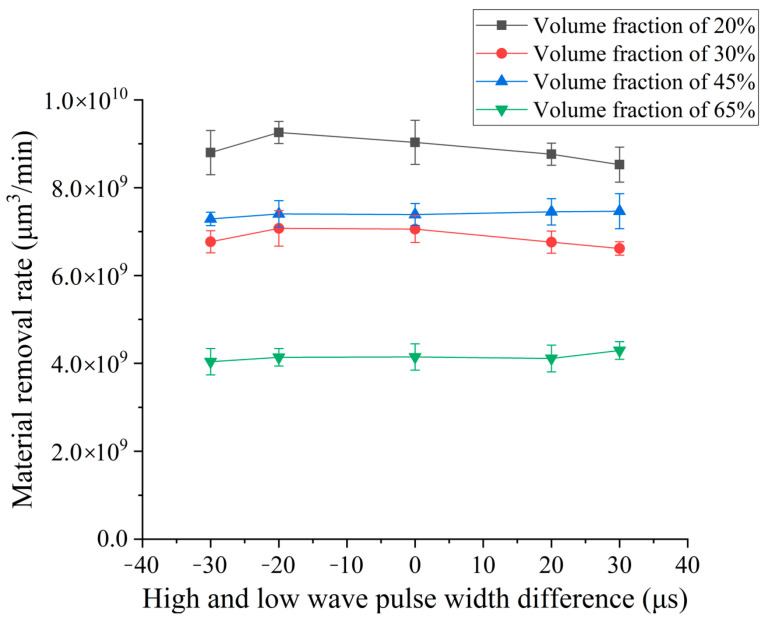
Effect of pulsed width difference on removal.

**Figure 6 micromachines-14-01315-f006:**
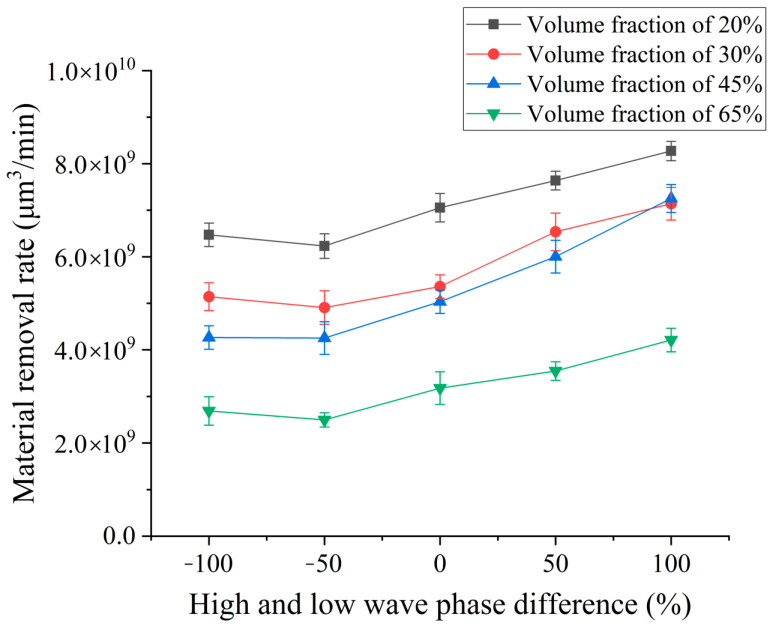
Effect of pulsed phase difference on removal.

**Figure 7 micromachines-14-01315-f007:**
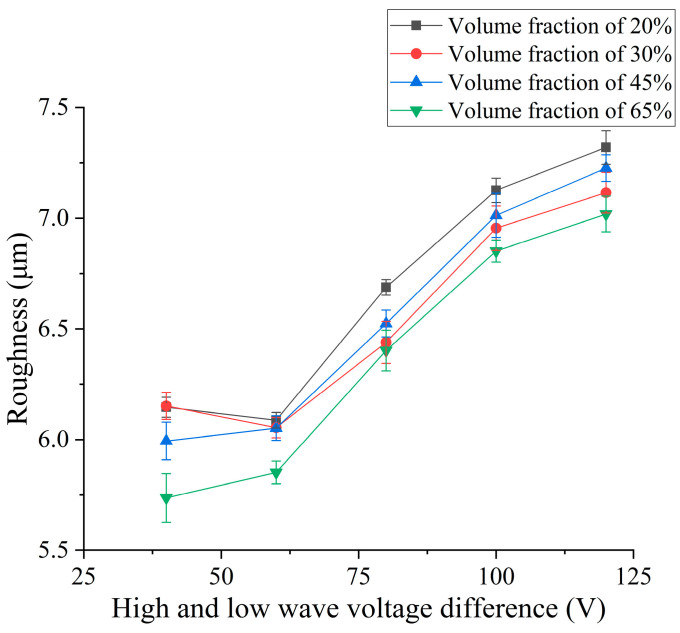
Effect of continuous pulsed open-circuit voltage difference on surface roughness.

**Figure 8 micromachines-14-01315-f008:**
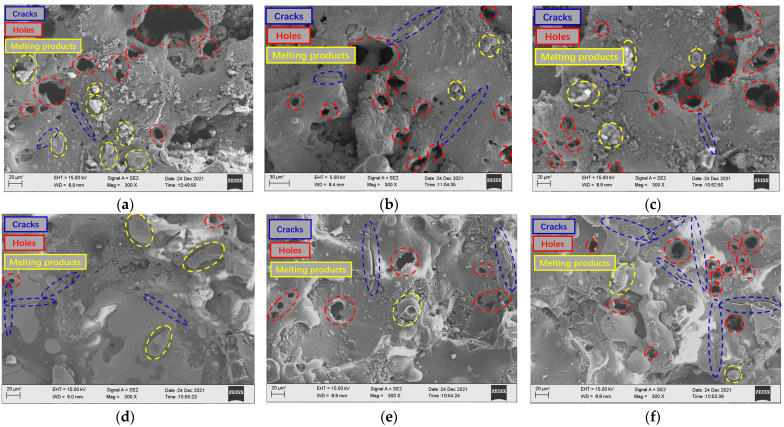
Surface morphology of high- and low-volume-fraction SiCp/Al composites (from top to bottom: low volume fraction to high volume fraction). (**a**) Open-circuit voltage difference 40 V. (**b**) Open-circuit voltage difference 80 V. (**c**) Open-circuit voltage difference 120 V. (**d**) Open-circuit voltage difference 40 V. (**e**) Open-circuit voltage difference 80 V. (**f**) Open-circuit voltage difference 120 V.

**Figure 9 micromachines-14-01315-f009:**
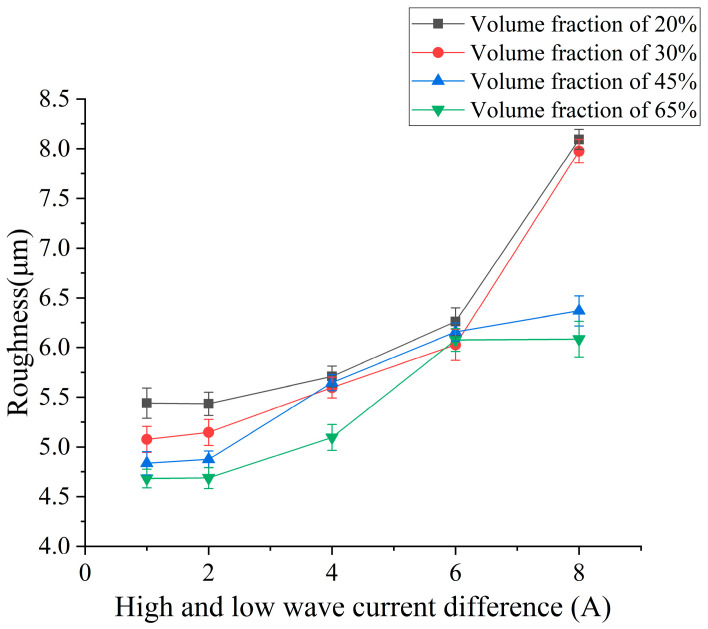
Effect of continuous pulsed peak current difference on surface roughness.

**Figure 10 micromachines-14-01315-f010:**
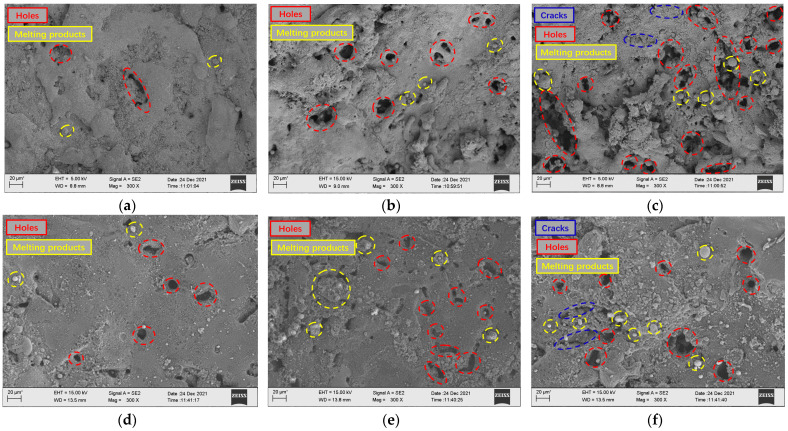
Surface morphology of low- and high-volume-fraction SiCp/Al composites (from top to bottom: low volume fraction to high volume fraction). (**a**) Peak current difference 1 A. (**b**) Peak current difference 4 A. (**c**) Peak current difference 8 A. (**d**) Peak current difference 1 A. (**e**) Peak current difference 4 A. (**f**) Peak current difference 8 A.

**Figure 11 micromachines-14-01315-f011:**
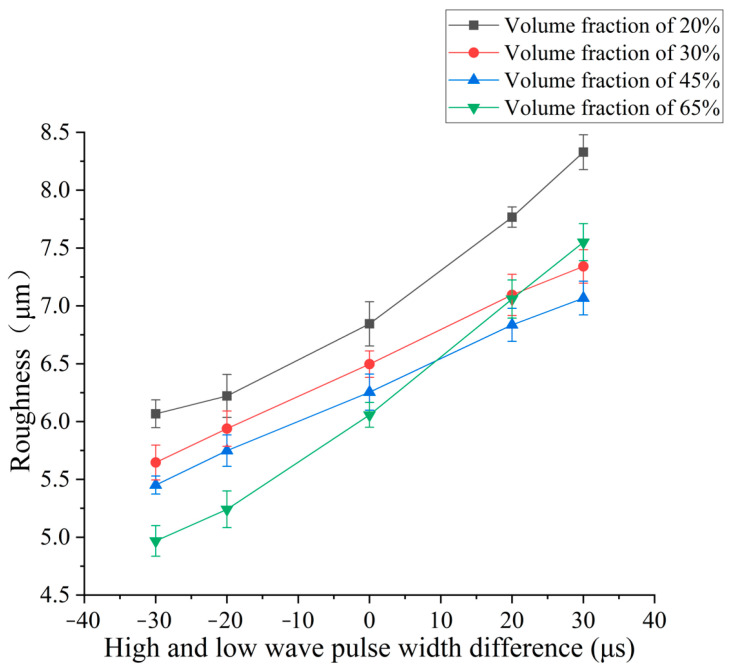
Effect of continuous pulse width difference on electrode loss rate.

**Figure 12 micromachines-14-01315-f012:**
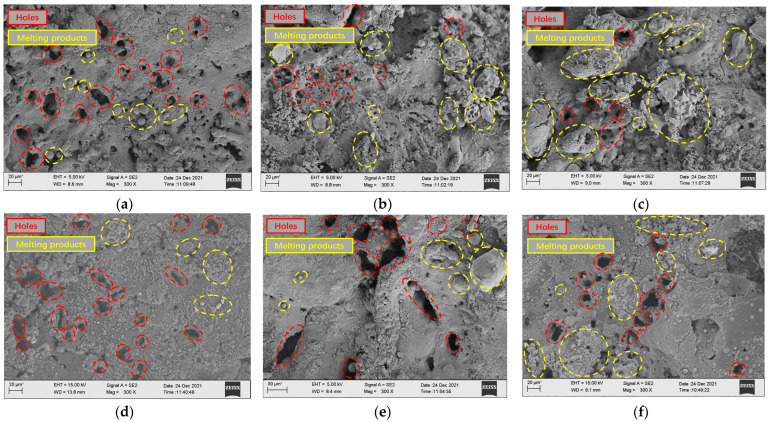
Surface morphology of low- and high-volume-fraction SiCp/Al composites (from top to bottom: low volume fraction to high volume fraction). (**a**) Pulse width difference −36 μs. (**b**) Pulse width difference 0 μs. (**c**) Pulse width difference 36 μs. (**d**) Pulse width difference −36 μs. (**e**) Pulse width difference 0 μs. (**f**) Pulse width difference 36 μs.

**Figure 13 micromachines-14-01315-f013:**
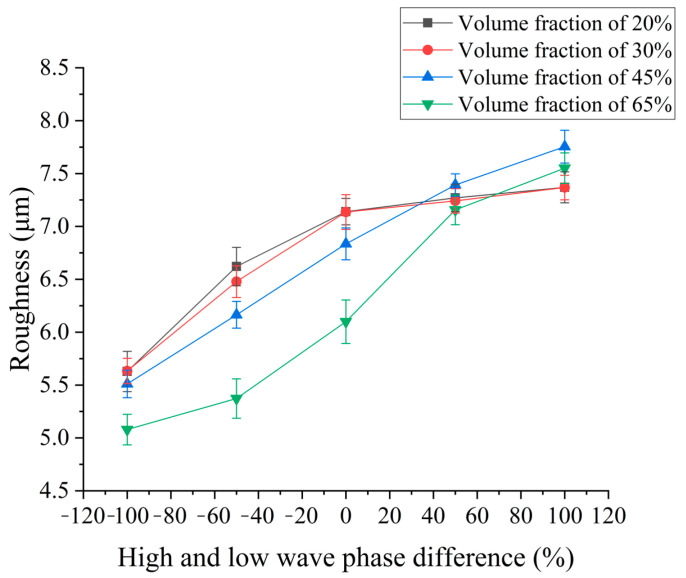
Effect of continuous pulse phase difference on the electrode loss rate.

**Figure 14 micromachines-14-01315-f014:**
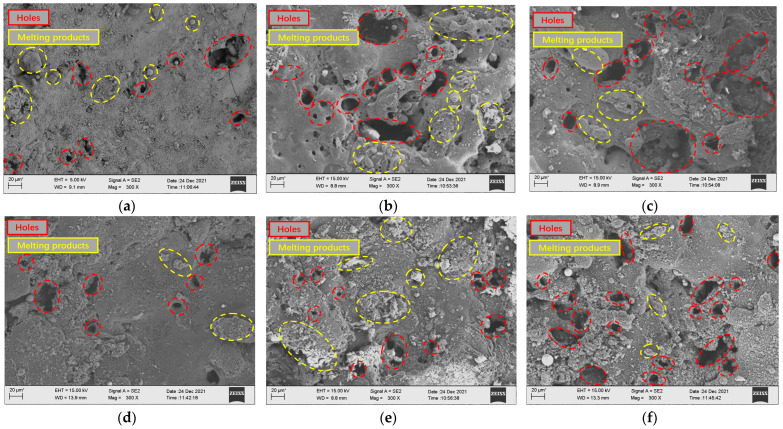
Surface morphology of low- and high-volume-fraction SiCp/Al composites (from top to bottom: low volume fraction to high volume fraction). (**a**) Pulse phase difference −100%. (**b**) Pulse phase difference 0%. (**c**) Pulse phase difference 100%. (**d**) Pulse phase difference −100%. (**e**) Pulse phase difference 0%. (**f**) Pulse phase difference 100%.

**Table 1 micromachines-14-01315-t001:** Comparison of performance indexes of SiCp/Al composites with different volume fractions.

Volume Fraction	Density g/cm^3^	Coefficient of Thermal Expansion (10^−6^/°C)	Thermal Conductivity (w/mk)
20%	2.82	14.9	170
30%	2.89	12.8	182
45%	2.92	11.0	195
65%	2.97	8.5	209

**Table 2 micromachines-14-01315-t002:** Main parameters of EDM machine.

Parameter	Specification
X, Y axis travel (mm)	195
Z-axis travel (mm)	120
Table size (mm)	240 × 300
Maximum feed speed (mm/min)	1200

**Table 3 micromachines-14-01315-t003:** Continuous pulse machining experiment parameters.

Experiment Number	High Wave Voltage (V)	High Wave Current (A)	High Wave Pulse Width, Low Wave Pulse Width (μs)	High Wave Phase, Low Wave Phase (%)	Low Wave Voltage (V)	Low Wave Current (A)
1	120, 140, 160, 180, 200	5	20, 20	0, 0	80	3
2	100	5, 6, 8, 10, 12	20, 20	0, 0	80	4
3	160	5	5, 35	0, 0	80	3
			10, 30			
			20, 20			
			30, 10			
			35, 5			
4	100	5	20, 20	−100, 100	80	3
				−50, 50		
				0, 0		
				50, −50		
				100, −100		

## Data Availability

The data used to support the findings of this study are available from the corresponding author upon request.

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
