# Peer review of "Study of the High-Efficiency Ejecting-Explosion EDM of SiCp/Al Composite"

_micromachines, 2023, doi:10.3390/mi14071315_

Round 1
Reviewer 1 Report
This paper investigates the effects of waveform parameters on the EDM machining of SiCp/Al composites. Some concerns need to be addressed first:
1. In Table 1, authors list the thermal expansion coefficient and thermal conductivity of the materials. How were they measured?
2. In Section 1.4, authors describe their methodology to determine the volume remove rate. A few examples are expected to evaluate the accuracy of the measurement.
3. More details about the experimental setup should be explained. For example, what is the geometry of the specimen? What is the speed of the machining? How long is the processing time? What is the machining pattern? Whether the specimens have been cut through? etc. To evaluate the effects of waveform parameters, other set up should be consistent.
4. Fig. 8, 10, 12 and 14 need some improvement. The layout is awful, the scale bars are hard to read, and the pink/yellow labels are low contrast on those SEM images (perhaps use other colors)
5. Since the analysis is based on the material removal rate and surface roughness at different DEM parameters, each parameter combination should be repeated at least three times to evaluate the standard deviation of the data points. Otherwise, it would be impossible to distinguish between the actual phenomena/tendency with the statistical fluctuations. Considering the number of parameter combinations, at least repeat those “anti-intrinsic” data points, such as the crossover curves, and abrupt increase/decrease.
No comment about the English language
Reviewer 2 Report
The author has well reported the high frequency ejecting-explosion EDM of SiCp/Al composite. However, minor revision is suggested to the author.
1. The manuscript has serious formatting issue including section heading, subheading, etc. (For example, line number 33) Many sentences are incomplete (For example, line number 113). The author is suggested to take careful revision of entire manuscript.
2. In Table 1, the unit of density and coefficient of thermal expansion are wrong. The author is suggested to take careful revision of entire manuscript to avoid such typo errors.
3. The reviewer is interested to know how the SiCp/Al composites of different volume fraction were synthesized as relevant information is missing from the manuscript.
4. From the Fig. 2, it is observed that the author has not used the standard EDM machine to perform the experiment. The reviewer is interested to know what is objective behind this? Further, the author is also suggested to add required information into the manuscript.
5. The reviewer is interested to know how the author has achieved the combination of high and low voltage waveform during EDM process as required information is missing from the manuscript.
6. In Table 3, the unit of voltage (v) is wrong, it should be (V).
7. In Fig. 9, Fig. 11 and Fig. 13, the author has mentioned the roughness value in micron. The reviewer is interested to know which roughness parameter was measured by the author?
8. The author is strongly advised to mention the future scope of the present study.
9. The author is strongly advised to mentioned the limitation of the current research work in the conclusion section.
Minor editing of English language required as many sentences are incomplete. The author is also suggested to take careful review of entire manuscript to avoid any typo errors.
Round 2
Reviewer 1 Report
Before accepting for publication, it needs a few improvements in the figures:
Fig. 8, 10, 12 and 14 now look better, but still insufficient for academic publications.
For example, Fig. 8 consists of 6 SEM images. It is recommended to index each image: 8(a), 8(b), 8(c), 8(d), 8(e), 8(f). And place the index label on the upper left corner of the image.
Place 2 or 3 images in each row to make the figure not too lengthy.
Adjust the contrast/brightness of the SEM images to make them look consistent. (details would be lost if the images are too dark or too bright)
No comment
